# Polyploidy of semi-cloned embryos generated from parthenogenetic haploid embryonic stem cells

**Eishi Aizawa, Charles-Etienne Dumeau, Remo Freimann, Giulio Di Minin, Anton Wutz***

Institute of Molecular Health Sciences, Swiss Federal Institute of Technology, ETH Zurich, Zurich, Switzerland

* awutz@ethz.ch

## Abstract

In mammals, the fusion of two gametes, an oocyte and a spermatozoon, during fertilization forms a totipotent zygote. There has been no reported case of adult mammal development by natural parthenogenesis, in which embryos develop from unfertilized oocytes. The genome and epigenetic information of haploid gametes are crucial for mammalian development. Haploid embryonic stem cells (haESCs) can be established from uniparental blastocysts and possess only one set of chromosomes. Previous studies have shown that sperm or oocyte genome can be replaced by haESCs with or without manipulation of genomic imprinting for generation of mice. Recently, these remarkable semi-cloning methods have been applied for screening of key factors of mouse embryonic development. While haESCs have been applied as substitutes of gametic genomes, the fundamental mechanism how haESCs contribute to the genome of totipotent embryos is unclear. Here, we show the generation of fertile semi-cloned mice by injection of parthenogenetic haESCs (phaESCs) into oocytes after deletion of two differentially methylated regions (DMRs), the *IG*-DMR and *H19*-DMR. For characterizing the genome of semi-cloned embryos further, we establish ESC lines from semi-cloned blastocysts. We report that polyploid karyotypes are observed in semi-cloned ESCs (scESCs). Our results confirm that mitotically arrested phaESCs yield semi-cloned embryos and mice when the *IG*-DMR and *H19*-DMR are deleted. In addition, we highlight the occurrence of polyploidy that needs to be considered for further improving the development of semi-cloned embryos derived by haESC injection.

## Introduction

The genetic information of an oocyte and a spermatozoon is passed onto the offspring. Both maternal and paternal genomes are required for normal development of mammalian embryos. Conversely, uniparental embryos suffer developmental defects due to the imbalance of genomic imprinting [1]. Despite the importance of the gametic genome, the prerequisites for forming a totipotent zygote remain incompletely understood. In mice, previous studies have shown

---

**Data Availability Statement:** All relevant data are within the manuscript and its Supporting Information files.

**Funding:** AW, grant 31003A_152814/1 from the Swiss National Science Foundation, WEB page:

---

www.snf.ch The funders had no role in study design, data collection and analysis, decision to publish, or preparation of the manuscript.

**Competing interests:** The authors have declared that no competing interests exist.

that two differentially methylated regions within the *H19-Igf2* and *Gtl2-Dlk1* imprinted gene clusters had critical contributions to the genome of embryos [2, 3]. Both regions are normally methylated and unmethylated on the paternally and maternally inherited chromosomes, respectively. Deletion of the *H19*-DMR and the *Gtl2-Dlk1* intergenic germline-derived DMR (*IG*-DMR) resulted in loss of expression of *H19* and *Gtl2* from the maternal allele, respectively [4, 5]. Deletion of the *H19*-DMR or combined deletions of the *H19*- and *IG*-DMRs from the genome of non-growing oocytes facilitated the generation of bimaternal mice after injection into mature oocytes [2, 3]. These studies defined a strategy for substituting sperm with a maternally derived genome. Recent studies have also explored the possibility to substitute the gametic genome by mouse haploid embryonic stem cells (haESCs). Haploid ESCs are unique stem cell lines established from either parthenogenetic [6, 7] or androgenetic haploid blastocysts [8, 9]. Haploid ESCs possess a single set of chromosomes, which contains 20 chromosomes in mice, similar to gametes. Due to their unique haploidy, haESCs have been applied in recent studies in original ways. One application is in genetic screening. While heterozygous mutations in diploid cells are often masked phenotypically, hemizygous mutations in haploid cells directly express phenotypes. For example, gene trap vectors have been used to screen genes required for chemical toxicity, self-renewal of ESCs, and X-chromosome inactivation [6, 10–12]. Another considerable application of haESCs is based on the similarity of their haploid genome to a gametic genome. Several reports have demonstrated the possibility for substituting the paternally derived sperm genome by either androgenetic haESC (ahaESC) or parthenogenetic haESC (phaESC) to generate semi-cloned mice [8, 9, 13, 14]. Furthermore, it has been demonstrated that the oocyte genome can be replaced by the genome of phaESCs for generation of semi-cloned mice, albeit at low frequency [15]. In contrast to oocytes and spermatozoa, genetic mutations can be readily introduced in haESCs owing to their self-renewal capacity in culture. Methods for introducing genetic modifications into the germline with haESCs is a considerable approach for studying embryonic development and for generating transgenic animals. Recent studies have also combined CRISPR-Cas9-based genome editing with haESCs for genetic screening [16], characterization of imprinting regions for embryonic development [17], and identification of important amino acids within the DND1 protein for primordial germ cell development [18].

While these remarkable studies have successfully applied haESCs as substitutes for gametic genomes, the mechanism how haESCs contribute to the genome of totipotent embryos remains to be clarified. For example, sperm and haESCs genomes are fundamentally different as most of the sperm genome is packaged with protamines, but the chromosomes of haESCs have a conventional nucleosomal structure. Proper segregation of both maternal and paternal haploid chromosome sets into each blastomere is required at the first division of the zygote to form a developmentally competent 2-cell embryo. Otherwise, developmental defects arise in embryos due to aneuploidy or polyploidy [19]. Polyploidy describes genomes with more than two complete sets of chromosomes, and is observed in several species including plants and yeasts [20]. In mammals, polyploid embryos can occur by polyspermy or abnormal chromosome segregation, but show developmental defects and arrest [20–24].

In this study, we report the generation of healthy mice by injection of mitotically arrested phaESCs that carry deletions of the *H19*- and *IG*-DMRs into metaphase II (MII) oocytes. We established semi-cloned ESCs (scESCs) from semi-cloned blastocysts to further characterize their genome. We find that several scESCs exhibited polyploidy, indicating that cautious analysis is required for the study of semi-cloned embryos generated by injection of haESCs.

## Results

### Deletion of the *IG*-DMR and *H19*-DMR in haESC lines

A previous study has reported that bimaternal embryos generated by substituting the paternal genome of sperm by the haploid genome of non-growing oocytes show developmental defects and arrest in embryogenesis [25]. These defects were largely overcome by manipulation of genomic imprinting. Deletion of the *IG*- and *H19*-DMRs from the genome of non-growing oocytes resulted in the development of bimaternal mice [2, 3]. These studies indicate that imprinted gene expression regulated by the *IG*-DMR and *H19*-DMR is the key barrier, which prevents the development of bimaternal embryos.

In order to manipulate genomic imprinting in phaESCs, the CRISPR-Cas9 system was used to delete the *IG*-DMR and *H19*-DMR in a phaESC line that was established from a 129S6/SvEvTac mouse oocyte (Fig 1A and S1A and S1B Fig). After transfection with expression vectors for CRISPR-Cas9 nucleases and guide RNAs, a *piggyBac* transposon plasmid for EGFP expression, and an expression vector for a *piggyBac* transposase, single cells were plated into multi-well dishes to establish clonal cultures. 2 double-knockout phaESC lines, DKO-phaESC-1 and DKO-phaESC-2, were identified by PCR-based genotyping (S1C Fig). These DKO-phaESC lines also expressed EGFP that allowed analysis of their contribution to embryos in further studies. DNA sequencing confirmed the deletions of 4,168 base pairs (bp) in the *IG*-DMRs for both DKO-phaESC-1 and DKO-phaESC-2 (Fig 1B). Similarly, deletions of 3,908 and 3,927 bp in the *H19*-DMRs were confirmed in DKO-phaESC-1 and DKO-phaESC-2, respectively. Both ESC lines showed a typical ESC morphology comparable to that of the parental phaESC line (Fig 1C). An intact haploid karyotype was confirmed by analysis of metaphase chromosome spreads (Fig 1D).

The maternally expressed *Gtl2* gene maps to a large imprinted cluster on mouse chromosome 12 and is regulated by the paternally methylated *IG*-DMR [26]. The maternally expressed gene *H19* maps close to the paternally expressed *Igf2* gene on chromosome 7 and is regulated by a shared *H19*-DMR. As expected, transcription of *Gtl2* and *H19* was lost in both DKO-phaESC lines (Fig 1E). In addition, expression of the paternally expressed *Dlk1* gene was slightly reduced in DKO-phaESC-1 and DKO-phaESC-2 compared with the expression of the parental phaESC line. No significant difference was observed in the expression of the paternally expressed *Igf2* gene.

### Generation of semi-cloned embryos and scESCs by injection of DKO-phaESCs into oocytes

For assessing the potential of DKO-phaESCs as sperm replacement, we injected single cells into MII oocytes, which were obtained from B6D2F1 females after superovulation (Fig 2A). Previously, ahaESCs that were arrested in mitosis at M-phase by demecolcine treatment had been used as sperm substitute with greater efficiency than ahaESCs in G0- or G1-phase [9]. Demecolcine binds to non-polymerized tubulin, and inhibits polymerization of microtubules leading to cell cycle arrest in mitosis without the formation of a spindle [27]. We treated DKO-phaESC-2 with demecolcine and purified the metaphase arrested 2n population by cell sorting (Fig 2B). Semi-cloned embryos were then constructed by injection of M-phase DKO-phaESC-2 cells into MII oocytes, followed by activation with strontium chloride. After activation, the majority of semi-cloned embryos exhibited weak EGFP fluorescence distributed over the cytoplasm. In a small number of embryos, a single round area of intense EGFP fluorescence was evident (Fig 2C), which indicated that the plasma membrane of DKO-phaESCs had inadvertently remained intact during injection. Semi-cloned embryos were subsequently cultured *in*

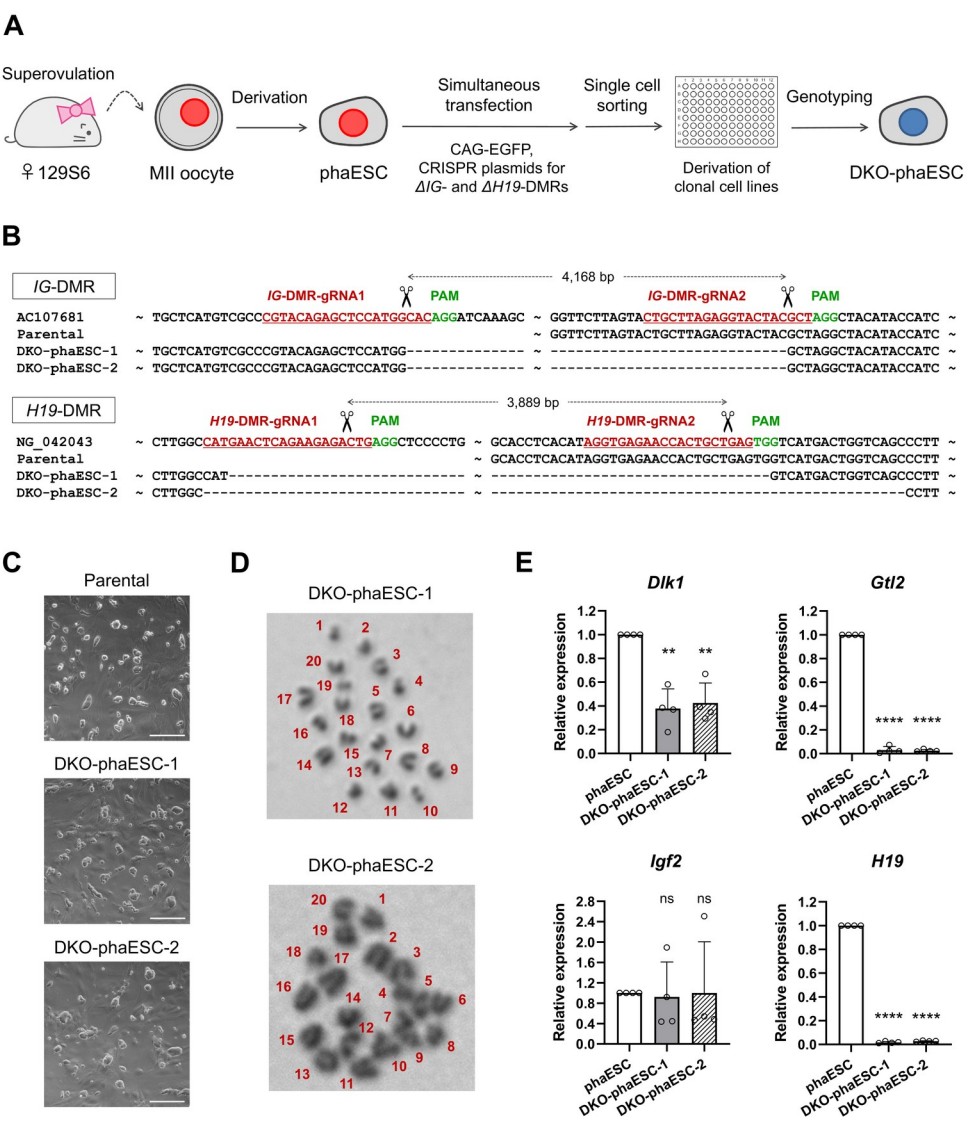

**Fig 1. Generation of the *IG*-DMR and *H19*-DMR deletions in haESCs.** (A) *IG*-DMR and *H19*-DMR deletions were engineered in phaESCs, which were established from haploid blastocysts obtained from activated mouse oocytes, by simultaneous transfection with four vectors encoding CRISPR-Cas9 nucleases, a CAG-EGFP-IRES-hygro *piggyBac* transposon vector, and a transposase vector. (B) Sequences of PCR fragments amplified over the deleted regions confirmed the loss of both DMRs in DKO-phaESC-1 and DKO-phaESC-2. (C) Morphology of DKO-phaESC lines. Scale bar, 200 μm. (D) Haploid karyotypes were observed in both DKO-phaESC-1 and DKO-phaESC-2. (E) Transcription of imprinted genes *Gtl2* and *H19*, both of which are maternally expressed and regulated by the *IG*- and *H19*-DMRs, was reduced in DKO-phaESC-1 and DKO-phaESC-2. Gene expression was normalized to *Gapdh* relative to the parental cell line. Data represents relative expression of each sample with the mean values and standard deviation (n = 4). **** $P < 0.0001$; ** $P < 0.01$; ns, non-significant.

*vitro* and developed to the 2-cell and blastocyst stage at the frequency of 58.7% (98/167) and 12.6% (21/167), respectively. At the 2-cell stage little or no EGFP expression was detected (Table 1). EGFP expression started gradually with the 4-cell stage at day 2 after injection. Finally, all the blastocysts exhibited EGFP expression, indicating DKO-phaESCs contributed to the blastocyst genome and no parthenogenetic blastocysts had developed.

To further analyze the semi-cloned embryos, we cultured 5 semi-cloned blastocysts and established 4 semi-cloned ESC lines, scESC-1 to scESC-4 (S2A Fig). Genotyping revealed that

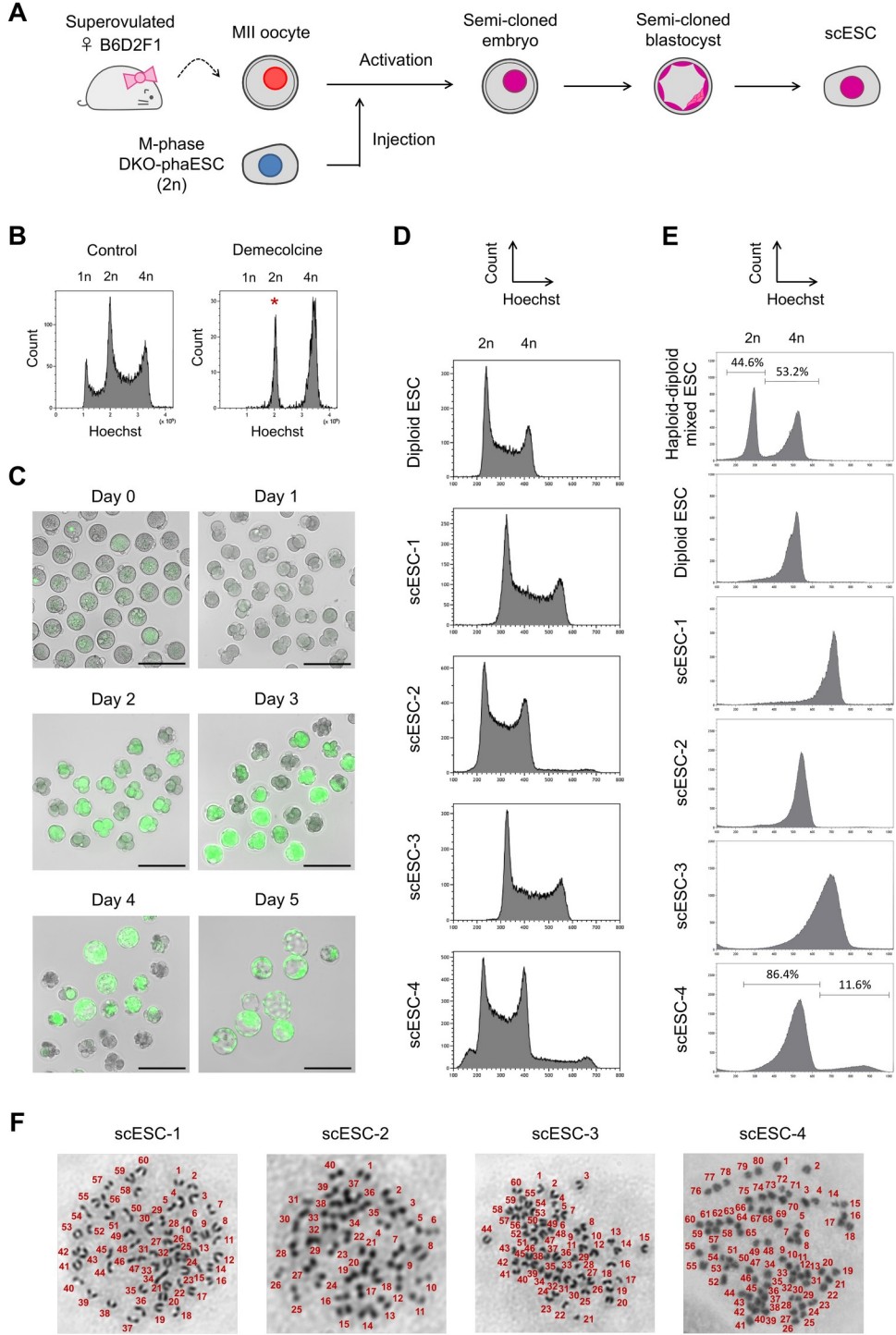

**Fig 2. Characterization of scESC lines derived by injection of DKO-phaESCs into oocytes.** (A) A scheme of the generation of scESC lines by injection of DKO-phaESCs into MII oocytes. (B) DKO-phaESCs were arrested in metaphase with demecolcine for 8 hours and sorted for a 2n DNA content. The peak of the Hoechst intensity corresponding to 2n DKO-phaESCs is indicated (asterisk). (C) Semi-cloned embryo development after injection of DKO-phaESCs into oocytes. EGFP fluorescence merged with bright field images are shown. At day 4, morulae developed to blastocysts. Scale bar, 200 μm. (D, E) DNA content analysis of 4 scESC lines, which were either untreated (D) or treated with demecolcine (E), by flow cytometry after Hoechst staining. (D) DNA content at the G1 phase of scESC-1 and scESC-3 appeared in the middle between the DNA content of G1 and G2 phase control diploid ESCs, indicating scESC-1 and scESC-3 are triploid. scESC-4 contained both diploid and tetraploid cells. (E) Only one

population of DNA content was observed in each scESC-1, scESC-2 and scESC-3, while scESC-4 showed two populations of different DNA contents. A haploid-diploid mixed ESC line and a diploid ESC line were included as a reference (top). The percentage of cells within the peaks is indicated in the histograms (top and bottom). (F) Metaphase spreads show triploid karyotypes of scESC-1 and scESC-3, and a tetraploid karyotype of scESC-4.

these scESC lines possessed both wild-type and deleted alleles of the *IG*-DMR and *H19*-DMR, confirming the contribution of DKO-phaESC and oocyte genomes (S2B Fig). We next analyzed the DNA content of all 4 scESC lines by flow cytometry after Hoechst staining (Fig 2D). In addition, the DNA content of M-phase arrested scESC lines was analyzed after the treatment with demecolcine for 8 hours (Fig 2E). scESC-2 exhibited an expected diploid DNA content, while the other 3 scESC lines appeared to be polyploid. All cells in scESC-1 and scESC-3 showed a triploid DNA content. scESC-4 contained cells with a diploid and tetraploid DNA content at the ratio of 86.4% and 11.6%, respectively. The analysis of metaphase chromosomes confirmed a triploid karyotype in scESC-1 and scESC-3, and a tetraploid karyotype in scESC-4 (Fig 2F). Considering that polyploidy is not compatible with mouse development [21–24], this observation might be relevant for improving semi-cloning by phaESC injection.

## Generation of semi-cloned mice from semi-cloned embryos

For further analysis of semi-cloned embryos, we performed embryo transfer to obtain semi-cloned mice. Constructed semi-cloned embryos were cultured to the 2-cell stage and transferred to oviducts of pseudopregnant Swiss Webster females. We chose Swiss Webster recipients for their albino coat color, which is readily distinguished from the agouti coat color of phaESCs and B6D2F1 oocytes. In parallel, albino 2-cell embryos were derived from Swiss Webster mice by *in vitro* fertilization (IVF) as a technical control. A total of 39 semi-cloned and 20 control 2-cell embryos were transferred to 4 recipient females (Table 2). Two of the four recipient females maintained pregnancy and delivered 6 pups (termed F0 no.1-6) and 1 pup (F0 no.7) (Fig 3A). F0 no. 6 and 7 had dark eye pigmentation and toe biopsies indicated EGFP expression under UV illumination (Fig 3B). Genotyping confirmed that F0 no. 6 and 7 were female and heterozygous for the *IG*-DMR and *H19*-DMR, carrying wild-type and deletion alleles (Fig 3C). Both mice grew normally without any apparent phenotypes or health problems. Furthermore, they were fertile and delivered full-term F1 pups, when mated with Swiss Webster males (Fig 3D). Transmission of the EGFP transgene was observed in about half of these F1 mice (7/15) in the expected Mendelian ratio (S3 Fig). Bisulfite DNA sequencing demonstrated that F0 no. 6 and 7 carried both methylated and unmethylated DMR alleles

**Table 1. Summary of preimplantation development of semi-cloned embryos derived by injection of DKO-phaESCs into oocytes.**

| | No. of oocytes injected | No. of 2-cell embryos | No. of 4-cell embryos | No. of morulae | No. of blastocysts (% of oocytes injected) |
|---|---|---|---|---|---|
| All oocytes or embryos | 167 | 98 | 50 | 43 | 21 (12.6%) |
| Embryos with EGFP expression | - | 0 | 31 | 34 | 21 |

**Table 2. Summary of semi-cloned mice generated by injection of DKO-phaESCs into oocytes.**

| Embryo types | No. of oocytes injected | No. of 2-cell embryos | No. of transferred 2-cell embryos | No. of delivered pups (% of transferred 2-cell embryos) |
|---|---|---|---|---|
| Control | - | - | 20 | 5 (25%) |
| Semi-cloned | 50 | 39 | 39 | 2 (5.1%) |

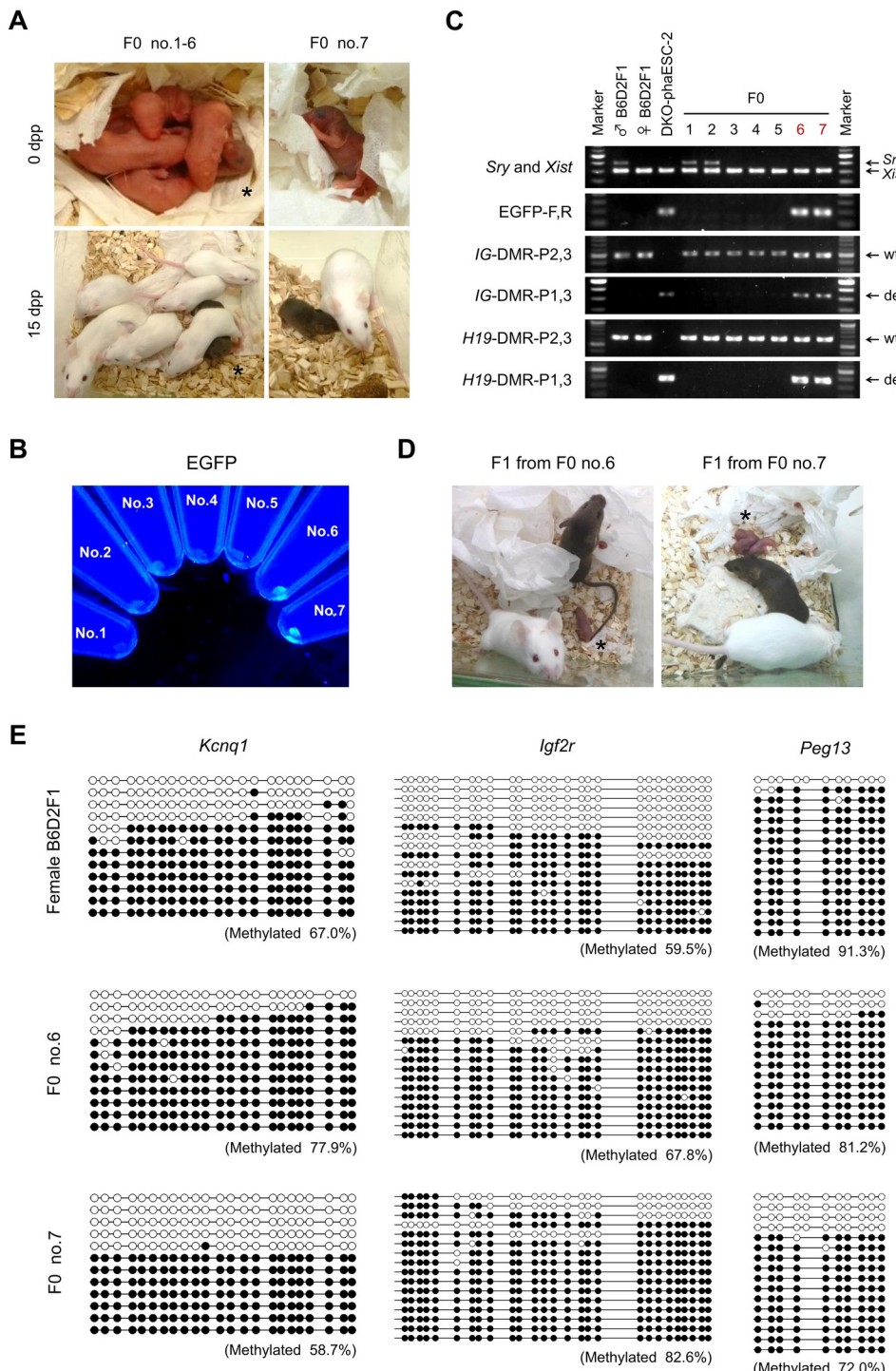

**Fig 3. Generation of semi-cloned mice by transfer of semi-cloned embryos into recipient mothers.** (A) 7 offspring (F0 no. 1–7) were obtained from 2 albino recipient mothers after transfer of semi-cloned and albino control 2-cell embryos. F0 no. 6 (indicated by asterisk) and no. 7 displayed black eyes and agouti coat color indicating DKO-phaESC derived pigmentation. (B) Toe biopsies of F0 no. 1–7. Biopsies of F0 no. 6 and 7 expressed EGFP under UV illumination. (C) Genotyping of F0 no. 1–7. F0 no. 6 and 7 possessed wild type and deletion alleles of the *IG*-DMR and *H19*-DMR. (D) Mating of semi-cloned F0 no. 6 and 7 with wild type males yielded healthy F1 pups (indicated by asterisk). (E) Bisulfite DNA methylation analysis of *Kcnq1*, *Igf2r* and *Peg13* in biopsies of F0 and a control mouse. White circles represent unmethylated CpGs; black circles represent methylated CpGs. The ratio of methylated CpGs is shown in brackets.

of the 3 imprinted genes *Kcnq1*, *Igf2r* and *Peg13* (Fig 3E). *Kcnq1* of F0 no.6, and *Igf2r* of F0 no. 6 and 7 appeared to be slightly hypermethylated compared to the control B6D2F1 female mouse. Conversely, *Peg13* was less methylated in F0 no. 6 and 7 than in the control. Considering that we observed slightly increased methylation levels in the control, which was comparable to that in F0 no. 6 and no. 7, we suggest that methylation of the 3 imprinted genes that we investigated was within a normal range in the 2 semi-cloned mice. This view is consistent with our observation that the semi-cloned mice were healthy and fertile.

## Discussion

The successful production of semi-cloned mice in our study shows that phaESCs with deletions of the *IG*-DMR and *H19*-DMR can replace sperm in mouse development. We unexpectedly observed that some scESC lines that we derived from semi-cloned embryos were triploid or contained a mixture of diploid and tetraploid cells (Fig 2D–2F). Several reasons for polyploid genomes in semi-cloned embryos can be considered.

One possible cause is the failure to extrude one haploid genome either from the spindle of the MII oocyte or of the M-phase arrested DKO-phaESC after the construction of a semi-cloned embryo. By following a previous study [9], we used DKO-phaESCs arrested at M-phase by demecolcine treatment as donor cells for semi-cloned embryos. Since demecolcine inhibits the polymerization of microtubules, the formation of mitotic spindles was presumably blocked in DKO-phaESCs, when they were injected into MII oocytes. It is possible that erroneous assembly of a spindle or erroneous attachment of the M-phase chromosomes of DKO-phaESC to the spindle after injection into oocytes might have caused chromosome segregation defects. Polyploidy could also arise, if long treatment with demecolcine led to diploidization of DKO-phaESCs. It is conceivable that some DKO-phaESCs might exit mitotic arrest and enter G1 phase without chromosome segregation and cytokinesis. However, we deem this an unlikely scenario. Lastly, the injection procedure might have induced errors in the MII spindle of the oocyte. In these three scenarios, 2n chromatids of either an MII oocyte or a DKO-phaESC would contribute to the embryo, resulting in a triploid genome.

A mixed karyotype of diploidy and tetraploidy was also observed in one scESC line. This is more difficult to explain. A mixed karyotype could possibly arise by erroneous chromosome segregation at the 2nd or a later cleavage division [28]. Alternatively, tetraploidy could have arisen during or after scESC derivation. However, we think this is unlikely considering that mouse ESCs stably maintain a normal diploid genome and the injection procedure would hardly affect ESC derivation, which was initiated 4 days later. We favor the interpretation that both diploid and tetraploid cells might have developed in the semi-cloned embryo. Several studies on chimeric blastocysts containing diploid and tetraploid embryonic cells have shown that tetraploid cells contributed to extra-embryonic tissues but rarely to the fetus [29–31]. Nevertheless, there has been a report that tetraploid embryos developed to blastocysts and formed fetuses after implantation [22]. Considering these results, it is possible that tetraploid cells developed to inner cell mass cells of the blastocyst, and both diploid and tetraploid scESCs were derived in one out of 4 lines in our study. The unexpected observation of polyploidy in several scESC lines is possibly an impediment for the development of semi-cloned mice and could be a target for improvements to increase the yield of normal semi-cloned embryos in the future [21–24]. Further studies are expected to reveal the mechanism of polyploidy in semi-cloned embryos.

The application of haESC for replacing gametic genomes has potential for genetics because mutations can be efficiently introduced into haESCs in contrast to oocytes and spermatozoa. We demonstrate that 2 DMRs can be deleted in a single step in haESCs with maintenance of a

haploid karyotype. The generation of transgenic embryos or mice by substituting haESCs for gametic genomes might be especially useful for allele specific analyses and for studying genomic imprinting. haESCs can be a tool for genetic screening of factors required for fertilization or embryogenesis through injection of genetically modified haESCs into oocytes. To date, remarkable studies have reported the application of this haESC technology for genetic screening [16–18]. The mechanism by which haESCs contribute to semi-cloned embryos remains an important focus of further investigation. The observation of polyploidy of semi-cloned embryos in our study emphasizes that further mechanistic insight into the contribution of haESC to semi-cloned embryos is needed for a better understanding of gametic genome adaptation and for increasing the efficiency of semi-cloning.

## Materials and methods

### Animals and experiments

C57BL/6J and DBA/2J mice were purchased from Charles River Laboratories (Wilmington, USA). Swiss Webster and 129S6/SvEvTac mice were purchased from Taconic Biosciences (Rensselaer, USA). All the mice were housed in the animal facility of ETH Zurich. All animal experiments were performed under the license ZH152/17 in accordance with the standards and regulations of the Cantonal Ethics Commission Zurich.

### Oocyte collection

Four- to five-week-old female mice were induced to superovulate by injection of 5 IU pregnant mare's serum gonadotropin followed by 5 IU human chorionic gonadotropin (hCG). Cumulus-oocyte complexes (COCs) were collected from the oviducts 15–17 hours after hCG injection and were placed in M2 medium. COCs were treated with 0.1% hyaluronidase until the cumulus cells dispersed.

### Derivation and culture of phaESC lines

Derivation of phaESC lines from 129S6/SvEvTac mice was performed as previously described [7]. For introducing deletions of the *IG*-DMR and *H19*-DMR using the CRISPR-Cas9 system, previously published oligonucleotides for guide RNAs (gRNAs) [16] were ligated into the pX330-U6-Chimeric_BB-CBh-hSpCas9 vector (Addgene, #42230) that was digested with BbsI restriction enzyme (S1 Fig). Sequences of gRNAs are listed in S1 Table. Simultaneous transfection of 4 Cas9/gRNA vectors, a *piggyBac* plasmid carrying a CAG-EGFP-IRES-hygro transgene, and a hyperactive *piggyBac* transposase plasmid was performed into a phaESC line using lipofectamine 2000 by following a manufacturer's protocol. Subsequently single EGFP expressing haploid cells were isolated by flow cytometer (MoFlo Astrios EQ, Beckman Coulter) after staining with 15 μg/ml Hoechst 33342 (Invitrogen) and cultured with irradiated mouse embryonic fibroblasts (MEFs) in 2i plus LIF medium [32, 33]. After the growth of clonal single colonies, a subset of cells in each line was analyzed by flow cytometer after staining with Hoechst to select cell lines containing haploid cells and were genotyped to screen cell lines with deletions of the *IG*-DMR and *H19*-DMR. Subsequently, haploid 1n cell population in each selected haploid cell line was purified by cell sorting after Hoechst staining and was cultured on a gelatin-coated plate without MEFs in 2i plus LIF medium. Each haploid cell line was maintained with purification of haploid cell population by cell sorting after Hoechst staining every 4–6 passages. Chromosome counting and the second genotyping of each cell line was performed after MEFs were excluded by cell passages.

## Construction of semi-cloned embryos

Construction of semi-cloned embryos was performed following a published protocol [16] with a few modifications. DKO-phaESCs were maintained without MEFs in 2i plus LIF medium [32, 33]. M-phase arrest was performed for DKO-phaESCs by culturing in the medium supplemented with 0.05 mg/ml demecolcine (Merck) for 8 hours. After staining with Hoechst, DKO-phaESCs with a 2n DNA content were sorted by flow cytometer. Sorted DKO-phaESCs were maintained in 2i plus LIF medium supplemented with 20 mM HEPES (Invitrogen) and the tube containing cell suspension was kept on ice until the use for injection. In parallel, MII oocytes were harvested from superovulated B6D2F1 females. To construct semi-cloned embryos, sorted single DKO-phaESCs were injected into MII oocytes using a piezo-driven micromanipulator (Eclipse Ti, Nikon; PiezoXpert, Eppendorf). After injection embryos were cultured in KSOM medium for 1 hour and subsequently activated for 6 hours in KSOM medium containing 5 mM strontium chloride and 2 mM EGTA. After activation, embryos were washed and cultured in KSOM medium at 37˚C under 5% $CO_2$ in air.

## Derivation of scESC lines

After the culture of semi-cloned embryos in KSOM medium for 4–5 days, each blastocyst was transferred on MEFs in serum plus LIF medium, which component was previously described [34]. For unhatched blastocysts, zona pellucida was removed by the treatment with the Tyrode's solution (Merck) before the transfer. After the expansion of blastocyst outgrowth, cells were maintained on MEFs until the passage 5. After the passage 5, scESCs were maintained on gelatin-coated plates without MEFs in serum plus LIF medium. Karyotyping by flow cytometry and chromosome counting was performed for scESCs at the passage 9.

## Genotyping

DNA extraction from cells and biopsies was performed using lysis buffer (100 mM Tris pH 8.5, 200 mM NaCl, 5 mM EDTA and 0.2% SDS) supplemented with 0.1 mg/ml proteinase K at 55˚C for at least 4 hours. Debris were pelleted by centrifuging for 5 minutes at 13,000 rpm. Supernatant was replaced into a new tube containing equal volume of isopropanol. After mixing, the tube was centrifuged for 5 minutes at 13,000 rpm to pellet precipitated genomic DNA. The pellet was washed with 70% ethanol and resuspended by 50–200 μl water. PCR was performed using Phusion Hot Start II DNA Polymerase (Thermo Fisher Scientific) following the manufacturer's protocol. PCR products were separated by electrophoresis on 1.5% agarose gels and stained with ethidium bromide for visualization under a UV transilluminator. Primers used for genotyping are listed in S1 Table.

## Transcription analysis

RNA was extracted using the RNeasy Mini Kit (Qiagen) following the manufacturer's protocol, including an on-column DNA digest using RNase-free DNase (Qiagen). RNA concentration was determined using a NanoDrop Lite (Thermo Fisher Scientific). 500 ng total RNA was reverse transcribed using the PrimeScript RT Master Mix (Takara) according to the manufacturer's instruction. RT-PCR was performed at a 384 well format on the 480 Lightcycler instrument (Roche) using KAPA SYBR FAST qPCR KIT (Kapa Biosystems). Fold change expression was calculated using the ΔΔct method. *Gapdh* expression was used for normalization. Primers used for transcription analysis are listed in S1 Table.

## Chromosome counting

For karyotyping, chromosome spreads of ESCs were prepared on glass slides as described [35]. Chromosomes were stained with Giemsa solution (Merck), washed with Gurr's buffer, and subsequently chromosomes were imaged under the microscope (Axio Observer Z1, Zeiss). Pictures were taken using an ORCA-Flash4.0 camera (Hamamatsu Photonics K.K.) and chromosome counts were determined.

## *In vitro* fertilization (IVF)

Sperm mass collected from the cauda epididymis of Swiss Webster males were pre-incubated in Sequential Fert (ORIGIO) at 37˚C under 5% $CO_2$ in air. COCs were harvested from the oviductal ampulla of superovulated Swiss Webster females. After 1 hour of pre-incubation of sperm mass, a small aliquot of sperm suspension was added to a Sequential Fert drop containing COCs. Six hours later, oocytes were washed and transferred to KSOM medium. Embryo development to the 2-cell stage was assessed after 24 hours of IVF.

## Embryo transfer

Recipient Swiss Webster females were mated with vasectomized Swiss Webster males the night before, and plugs were confirmed in the morning of the day of the embryo transfer. Nine or ten 2-cell embryos derived by DKO-phaESC injection and 5 control 2-cell embryos by IVF were transferred into the oviducts of pseudo-pregnant recipient females. On day 19.5 of gestation, full-term pups were naturally delivered from recipient females.

## Bisulfite sequencing

Genomic DNA was extracted from toes of newborn semi-cloned mice and ear biopsy of 3 weeks old B6D2F1 mice with lysis buffer containing proteinase K, followed by isopropanol precipitation. Bisulfite conversion was performed using the EZ DNA methylation Gold kit (ZYMO Research). PCR was performed under the following temperature profile: 30 sec 98˚C, 20 x (10 sec 98˚C, 30 sec 65–55˚C with -0.5˚C per cycle, 30 sec 72˚C), 35 x (10 sec 98˚C, 30 sec 55˚C, 30 sec 72˚C), 5 min 72˚C. The PCR products were cloned into pJet1.2 vector using the CloneJET PCR Cloning Kit (Thermo Fisher Scientific), followed by the transformation into competent DH5α *E. coli*. Insert sequences for each colony were obtained through the commercial Ecoli NightSeq service (Microsynth). Bisulfite sequencing was analyzed with the QUMA methylation analysis tool (http://quma.cdb.riken.jp/). Primers used for PCR and sequencing are listed in S1 Table.

## Statistical analysis

For comparison of quantitative RNA expression levels of imprinted genes, measurements were analyzed with the GraphPad Prism 8 software using a two-tailed unpaired t-test. A p-value < 0.05 was considered statistically significant.

## Supporting information

**S1 Fig. Deletions of the *IG*-DMR and *H19*-DMR in phaESC lines.** (A) A design of gRNAs and primers targeting the deletions of the *IG*-DMR. (B) A design of gRNAs and primers targeting the deletions of the *H19*-DMR. (C) PCR fragments flanking both *IG*-DMR (319 bp) and *H19*-DMR (407 bp) by primers targeting deleted loci were observed in 2 DKO-phaESC lines, whereas the deleted sequences were absent in DKO-phaESC-1 and DKO-phaESC-2. (PDF)

**S2 Fig. Derivation and genotyping of scESC lines.** (A) Derivation of scESC lines from blasto-cysts generated by injection of DKO-phaESCs into oocytes. Images of blastocysts, outgrowth (passage 0) and scESCs after derivation are shown. Regular black bar, 100 μm; bold black bar, 200 μm; white bar, 100 μm. (B) Genotyping of 3 scESC lines. All 3 scESC lines exhibited both wild type and mutant alleles for the *IG*-DMR and *H19*-DMR, indicating both oocytes and DKO-phaESCs genome contributed to the genome of blastocysts.
(PDF)

**S3 Fig. Genotyping of F1 generation born to semi-cloned F0 mice.** PCR-based genotyping was performed for 15 F1 mice born to semi-cloned females (F0 no.6 and 7) and wild type Swiss Webster males. EGFP transgene was inherited to 7 among 15 F1 mice.
(PDF)

**S1 Table. List of oligos.**
(PDF)

## Acknowledgments

We thank Mr. Stefan Butz and Dr. Tuncay Baubec for providing primers and advice on bisul-fite sequencing. We also acknowledge Ms. Michèle Schaffner and Mr. Thomas M. Hennek for their technical support on embryo transfer.

## Author Contributions

**Conceptualization:** Eishi Aizawa, Anton Wutz.

**Data curation:** Eishi Aizawa, Charles-Etienne Dumeau, Remo Freimann.

**Formal analysis:** Eishi Aizawa, Charles-Etienne Dumeau, Remo Freimann.

**Funding acquisition:** Anton Wutz.

**Investigation:** Eishi Aizawa, Charles-Etienne Dumeau.

**Methodology:** Eishi Aizawa, Charles-Etienne Dumeau, Remo Freimann, Giulio Di Minin.

**Project administration:** Eishi Aizawa, Anton Wutz.

**Resources:** Eishi Aizawa, Charles-Etienne Dumeau, Remo Freimann, Giulio Di Minin.

**Supervision:** Anton Wutz.

**Validation:** Eishi Aizawa, Charles-Etienne Dumeau.

**Visualization:** Eishi Aizawa, Charles-Etienne Dumeau.

**Writing – original draft:** Eishi Aizawa, Anton Wutz.

**Writing – review & editing:** Eishi Aizawa, Charles-Etienne Dumeau, Remo Freimann, Giulio Di Minin, Anton Wutz.

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
