## [Decision Letter · Decision Letter 0]

28 Jun 2020

PONE-D-20-12064

Polyploidy of semi-cloned embryos generated from parthenogenetic haploid embryonic stem cells

PLOS ONE

Dear Dr. Wutz,

Thank you for submitting your manuscript to PLOS ONE. After careful consideration, we feel that it has merit but does not fully meet PLOS ONE’s publication criteria as it currently stands. Therefore, we invite you to submit a revised version of the manuscript that addresses the points raised during the review process.

Please submit your revised manuscript within 3 months. If you will need more time than this to complete your revisions, please reply to this message or contact the journal office at plosone@plos.org. Please include the following items when submitting your revised manuscript:

We look forward to receiving your revised manuscript.

Kind regards,

Jon Schoorlemmer, PhD

Academic Editor

PLOS ONE

Journal Requirements:

Additional Editor Comments (if provided):

Dear Dr Wutz

Both reviewers agree that "The authors provide a comprehensive and complete manuscript supported by high technical standards, the conclusions drawn are appropriately and supported by the data".

However, both reviewers raise questions that need to be answered and/or addressed before the manuscript can be accepted.

Best regards

Jon Schoorlemmer

Reviewers' comments:

Reviewer's Responses to Questions

**Comments to the Author**

1. Is the manuscript technically sound, and do the data support the conclusions?

Reviewer #1: Yes

Reviewer #2: Yes

2. Has the statistical analysis been performed appropriately and rigorously? 

Reviewer #1: N/A

Reviewer #2: Yes

3. Have the authors made all data underlying the findings in their manuscript fully available?

Reviewer #1: Yes

Reviewer #2: Yes

4. Is the manuscript presented in an intelligible fashion and written in standard English?

Reviewer #1: Yes

Reviewer #2: Yes

5. Review Comments to the Author

Reviewer #1: This manuscript concerns the manipulation of haESC to eliminate DMRs; the generation of healthy mice by zygotic injection of mitotically arrested phaESCs with eliminated DMRs and supposed effects on posterior genomic imprinting; and the use of ESC from semi-cloned blastocyst to analyze ploidy. The highly demanding experiments and controls are well designed and carried out, the manuscript is well written, the Figures excellently explicative and the sections are well organized. I have a few comments for the authors.

1 Except for cloning experts (and maybe not even all of them), the use of demecolcine for metaphase arrest is not common knowledge. Its use and desired molecular effects can be better explained. Has ploidy of the positively-sorted cells been assessed post-sorting?; are the authors sure the cell suspension was 100% single cells? Information should be provided regarding the media used pre- and post-sorting.

The ploidy of the cells should be mentioned in line 147

2 Lines 243 - 245: the generation of semi-cloned mice is called efficient. How do the rates mentioned stack up against published reports?

3 It would be relevant to explain from how many semi-cloned blastocysts the ESC lines were derived.

Line 270: also frequency ?

Line 283: The frequency depends on the number of blastocysts from which these lines were derived

4 As a general point, conclusions could be toned down to reflect the limited number of animals, blastocysts and scESC lines, etc that were analyzed to reach them. For example lines 170 and 247.

Minor points

- Some numbers are required to support the following statement: "Some methylation patterns including Kcnq1 and Peg13 of 213 progeny no.6 and Igf2r of progeny no. 7 appeared slightly hypermethylated".

- It would be of interest to know about the methylation status in semi-cloned animals of the DMRs deleted from the haESC. Is information available from these loci? If not, can authors explain why these experiments were not carried out or not included?

- procedures and medium used for "Each haploid cell line was maintained without mouse 318 embryonic fibroblasts" should be mentioned briefly

- The failure of the oocyte to separate haploid genomes after metaphase arrest, appears an obvious mechanisms for the polyploidy observed. Could the authors comment?

- line 313 manufacture’s protocol

The manuscript could use some language editing. While most is very well written, some specific passages/paragraphs/lines are not (I cite some but not all examples):

-"because uniparental embryos cause developmental " is simply incorrect, defect should be replaced by "suffer".

-The genetic information of an oocyte and a spermatozoon are inherited to the offspring. I would suggest "are inherited" be replaced by "is passed onto", or "is inherited by".

- line 264 "retarded before 15 days of gestation" is not clear

- sentence in lines 237-239 could be rewritten for clarity

- lines 250 and 266 overlap

- lines 239/240 are a copy of lines 143-145

Reviewer #2: In this manuscript Aizawa and colleagues provide a characterization of embryonic stem cell (ESC) lines derived from semi-cloned embryos generated by injection of parthenogenetic haploid ESC (phaESC) into mouse oocytes.

The authors show here: 1) generation of phaESC injection carrying the double deletion of H19 and IG - paternal differentially methylated regions (DMRs), 2) generation of mouse competent semi-cloned embryos by phaESC doubleKO lines and characterization of ESC derived from blastocyst (termed scESC), 3) generation of F1 progeny and targeted DNA methylation analysis of imprinted genes.

The authors provide a comprehensive and complete manuscript, the experimental design seems supported by high technical standards, the drawn conclusions are appropriate and supported by the data.

Although the efficiency and a broad characterization of this line of approaches have been previously shown by this and others laboratories (e.g. requirement of deletion of IG and H19 paternally DMRs, successful generation of semi-cloned embryos by injection of haploid cells in MII oocytes), I do consider genuinely interesting and novel the discovery of frequent polyploidy on scESC, raising a potential explanation of the low developmental competence of embryos obtained by this methodology. Thus, I think that the results shown in this manuscript could be of interest for the scientific community.

However, I consider that some aspects should be clarified prior to its acceptance for publication in PLoS One.

Major points:

- Line 155: Is the Table 1 referring to the embryos that show eGFP expression, or the ones that are simply developing? Could you provide a more comprehensive table of the development of injected embryos including the percentage of which show eGFP expression? This information seems to be provided only at the blastocyst stage.

- Line 159: mESC cultured on 2i have shown an altered karyotype over culture passaging. How many passages do have scESC cell lines when analyzed? In material and methods is mentioned that purification of phaESC is performed every 4-6 passages but I could not find when chromosome countings of scESC are performed. Which culture media has been used? (I could not find it either on Materials and methods). Do you think that this scESC derivation could influence polyploidy of the cells?.

- Lines 167 - 172: It is interesting the raised hypothesis that polyploidy could be an important limiting factor in the embryonic development of semi-cloned embryos. The authors say that is frequent and prevalent, but how prevalent polyploidy is in scESC derived lines?. Could the authors provide the percentage of scESC that showed polyploidy by karyotype analysis in the different cell lines?.

- Line 194 - 196: This section introduction is rather confusing as semi-cloned embryos competence to develop to mice has been previously proved, as the authors have cited in Refs 13 and 14. Please rephrase this sentence.

- Lines 243 - 244: I find a bit confusing the way of presenting the percentage of efficiently generated semi-cloned blastocysts and mice in this sentence. Is this referring to the developmental progression shown in Figure 2 (blastocysts), or the one of semi-cloned mice shown in Figure 3?. Please clarify this issue.

Minor points

- Supplementary Fig2A and Fig2B, why there are images and genotyping controls of only 3 scESC lines?

- Line 95, ‘generaterd’

- Line 208 -210: Please refer to the subsequent progenies as F0 and F1.

6. PLOS authors have the option to publish the peer review history of their article (what does this mean?). If published, this will include your full peer review and any attached files.

Reviewer #1: No

Reviewer #2: No

---

## [Author Response · Author response to Decision Letter 0]

3 Aug 2020

Point to point response to the reviewers’ comments

Response to Reviewer #1

1. Except for cloning experts (and maybe not even all of them), the use of demecolcine for metaphase arrest is not common knowledge. Its use and desired molecular effects can be better explained. Has ploidy of the positively-sorted cells been assessed post-sorting?; are the authors sure the cell suspension was 100% single cells? Information should be provided regarding the media used pre- and post-sorting. The ploidy of the cells should be mentioned in line 147.

Response: We thank the reviewer for the suggestion and have added a brief explanation of the effect of demecolcine in lines 148-150. Demecolcine depolymerizes mircotubules and therefore prevents the formation of a mitotic spindle and causes a cell cycle arrest in M-phase. The haploid cell population arrested at M-phase, which were sorted by the flow cytometry, overlaps with the population of diploid cells in G0/G1-phase (Figure 2B). Considering that demecolcine gives a near complete arrest in mitosis and no 1n peak corresponding to a haploid G0/G1-phase population was observed after demecolcine treatment, diploid cells at G0/G1-phase can be excluded from the sort gate. A single haESC was then injected into each oocytes. Thereby, it was visibly confirmed that the injected cell was a single cell and not an aggregate or cluster of cells.

The information on medium used pre- and post-sorting has been added in lines 343-349 as requested.

In addition, we discuss now a potential role of demecolcine treatment for emergence of polyploidy of semi-cloned ESCs in lines 251-262.

2. Lines 243 - 245: the generation of semi-cloned mice is called efficient. How do the rates mentioned stack up against published reports?

Response: We thank the reviewer for bringing the unclear language to our attention. Our frequency for obtaining semi-cloned embryos is in line with the literature. However, considering limited numbers in our study, we have toned down the conclusions and have rephrased the text to refrain from making assessments of efficiency concerning our semi-cloning experiments. Consequently, we have deleted lines 243-245 of the original version in the revision.

3. It would be relevant to explain from how many semi-cloned blastocysts the ESC lines were derived.

Line 270: also frequency?

Line 283: The frequency depends on the number of blastocysts from which these lines were derived

Response: We have established 4 scESC lines from 5 semi-cloned blastocysts. This information has been added in lines 165-166 in the revised text. In addition, we have added an explanation in lines 291-293, that one of the scESC lines contained a mixture of diploid and tetraploid cells. In line 283, ‘frequent’ has been deleted (new line 295-298) due to the limited number of samples.

4. As a general point, conclusions could be toned down to reflect the limited number of animals, blastocysts and scESC lines, etc that were analyzed to reach them. For example lines 170 and 247.

Response: We have followed the reviewer’s suggestion and toned down our conclusions and removed the statements on efficiency. We have rewritten the sentence in line 170 of the original text (new lines 172-173). In addition, we have removed the discussion of cell cycle synchronization in former line 247.

Minor points

- Some numbers are required to support the following statement: "Some methylation patterns including Kcnq1 and Peg13 of 213 progeny no.6 and Igf2r of progeny no. 7 appeared slightly hypermethylated".

In response, we have added the percentage of methylated CpGs to the panel in Figure 3E and a description that more than 75% of methylated CpGs were observed over these 3 DMRs has been added in the text (lines 223-224).

- It would be of interest to know about the methylation status in semi-cloned animals of the DMRs deleted from the haESC. Is information available from these loci? If not, can authors explain why these experiments were not carried out or not included?

Response: The reviewer asks for the methylation status of the H19- and IG-DMRs. Since these DMRs are deleted in the DKO-phaESCs their methylation status cannot be assessed in the semi-cloned scESCs or mice. Only the oocyte-derived allele of these DMRs is present. As the maternal oocyte-derived DMRs were not manipulated in our experiments, we would strongly suggest that their methylation status is normal and, thus, predictable. We have therefore focused on the analysis of 3 imprinted genes (Kcnq1, Igf2r and Peg13), in which semi-cloned mice possess 2 alleles.

- procedures and medium used for "Each haploid cell line was maintained without mouse 318 embryonic fibroblasts" should be mentioned briefly

Response: In the revised version, details of the procedures and haESC culture have been added in lines 328 and 338.

- The failure of the oocyte to separate haploid genomes after metaphase arrest, appears an obvious mechanisms for the polyploidy observed. Could the authors comment?

Response: We thank the reviewer for the comment and agree that a failure in separating the oocyte haploid genomes is a possible cause of polyploidy. In addition, the failure to separate the sister chromatids of M-phase arrested DKO-phaESCs would similarly cause polyploidy. This could be a potential consequence of demecolcine treatment. We have now added these points to the discussion in lines 251-259. 

- line 313 manufacture’s protocol

Response: The word was revised as manufacturer’s protocol.

The manuscript could use some language editing. While most is very well written, some specific passages/paragraphs/lines are not (I cite some but not all examples):

-"because uniparental embryos cause developmental " is simply incorrect, defect should be replaced by "suffer".

Response: The sentence was changed to “because uniparental embryos suffer developmental defects” in line 37.

-The genetic information of an oocyte and a spermatozoon are inherited to the offspring. I would suggest "are inherited" be replaced by "is passed onto", or "is inherited by".

Response: The sentence was changed to “The genetic information of an oocyte and a spermatozoon is passed onto the offspring.” in line 35.

- line 264 "retarded before 15 days of gestation" is not clear

Response: The sentence was changed to “the most viable embryos developed until the 15th day of gestation” in lines 276-277.

- sentence in lines 237-239 could be rewritten for clarity

Response: We have rewritten the sentence (former lines 237-239; now in lines 253-254).

- lines 250 and 266 overlap

Response: We have removed the sentence in the former line 266.

- lines 239/240 are a copy of lines 143-145

Response: The sentences in the former lines 239/240 have been deleted.

Response to Reviewer #2

Major points:

- Line 155: Is the Table 1 referring to the embryos that show eGFP expression, or the ones that are simply developing? Could you provide a more comprehensive table of the development of injected embryos including the percentage of which show eGFP expression? This information seems to be provided only at the blastocyst stage.

Response: We have previously shown the development of all the embryos including embryos that did not show EGFP expression in Table 1. In the revision, we have added a row to the table that lists the number of embryos showing EGFP expression. We point out that GFP expression appeared gradually from the 4-cell stage on and it is likely that a detection threshold was reached by different embryos at different times. At the blastocyst stage all embryos showed GFP expression suggesting that heterogeneity at earlier stages was likely due to the timing of GFP expression or our ability to clearly detect green fluorescence under the microscope.

- Line 159: mESC cultured on 2i have shown an altered karyotype over culture passaging. How many passages do have scESC cell lines when analyzed? In material and methods is mentioned that purification of phaESC is performed every 4-6 passages but I could not find when chromosome countings of scESC are performed. Which culture media has been used? (I could not find it either on Materials and methods). Do you think that this scESC derivation could influence polyploidy of the cells?

Response: We thank the reviewer for pointing out this important detail. We are aware of reports of chromosomal instability of mouse ESCs in 2i medium from several labs. To exclude this as an effect we have established and maintained our scESC lines in serum plus LIF medium. This detail has been added in lines 357-365 of the methods section. DNA content analysis by flow cytometry and chromosome counting were performed at the passage 9. Considering scESC lines were karyotyped at early passage (passage 9), we suggest that culture did not have a material influence on polyploidy of scESC lines. This view is consistent with the observation that mouse ESCs rarely become polyploid in culture. Frequent aneuploidies are caused by the gain or loss of individual chromosomes including a gain of chromosome 8 and loss of one of one X chromosome in female ESCs.

- Lines 167 - 172: It is interesting the raised hypothesis that polyploidy could be an important limiting factor in the embryonic development of semi-cloned embryos. The authors say that is frequent and prevalent, but how prevalent polyploidy is in scESC derived lines?. Could the authors provide the percentage of scESC that showed polyploidy by karyotype analysis in the different cell lines?.

Response: In order to accurately assess the percentage of polyploid cells in each of our 4 scESC lines, we have measured their DNA content by flow cytometry after demecolcine treatment. This allows us to observe the cells arrested in mitosis and therefore each ploidy state can be assigned a single peak. From this analysis we find that the scESC-1, scESC-2, and scESC-3 are homogenous cultures, whereby scESC-2 is diploid, and scESC-1 and scESC-3 are triploid. scESC-4 is a mixed culture comprised of 11.6% tetraploid and 86.4% diploid cells. The new data is shown in Figure 2E and discussed in the text in lines 170-175.

- Line 194 - 196: This section introduction is rather confusing as semi-cloned embryos competence to develop to mice has been previously proved, as the authors have cited in Refs 13 and 14. Please rephrase this sentence.

Response: We have rephrased the sentence explaining the aim for further analysis of semi-cloned embryos in lines 205-206 following the reviewer’s suggestion.

- Lines 243 - 244: I find a bit confusing the way of presenting the percentage of efficiently generated semi-cloned blastocysts and mice in this sentence. Is this referring to the developmental progression shown in Figure 2 (blastocysts), or the one of semi-cloned mice shown in Figure 3?. Please clarify this issue.

Response: We thank the reviewer for raising this point. We have removed statements of efficiency regarding our semi-cloning experiments considering the limited numbers in our study. Therefore, we have removed the sentences in the former lines 243-244. Please, see also the response to reviewer 1 points 2 to 4.

Minor points

- Supplementary Fig2A and Fig2B, why there are images and genotyping controls of only 3 scESC lines?

Response: We have added images and genotyping results and now show the data for all 4 scESC lines in Supplementary Fig2A and 2B.

- Line 95, ‘generaterd’

Response: We have corrected the spelling mistake.

- Line 208 -210: Please refer to the subsequent progenies as F0 and F1.

Response: We thank the reviewer for the suggestion for clarifying the text. We have changed the text (line 212-223) and figures (Figure 3 and Supplementary Figure 3) and now refer to the semi-cloned mice as F0, their offspring as F1 generation to make the discussing clearer.

---

## [Editor Report · Decision Letter 1]

11 Aug 2020

PONE-D-20-12064R1

Polyploidy of semi-cloned embryos generated from parthenogenetic haploid embryonic stem cells

PLOS ONE

Dear Dr. Wutz,

Thank you for submitting your manuscript to PLOS ONE. After careful consideration, we feel that it has merit but does not fully meet PLOS ONE’s publication criteria as it currently stands. Therefore, we invite you to submit a revised version of the manuscript that addresses the points raised during the review process.

A rebuttal letter that responds to each point raised by the academic editor and reviewer(s). You should upload this letter as a separate file labeled 'Response to Reviewers'.An unmarked version of your revised paper without tracked changes. You should upload this as a separate file labeled 'Manuscript'.

We look forward to receiving your revised manuscript.

Kind regards,

Jon Schoorlemmer, PhD

Academic Editor

PLOS ONE

Additional Editor Comments (if provided):

Dear author, dear Dr Wutz,

As the authors suggest in the cover letter, most points that were raised by the reviewers have been addressed, with the inclusion of additional data and text clarifications.

Before formal acceptation however, a few more minor point should be adressed, one of which became clear only once methylation percentages were added to Figure 3E.

1 Lines 223-227.

The description of this Figure is partially incorrect, and the conclusion is still overstated:

- The sentence “patterns ………….appear hypermethylated” is gramatically incorrect.

- The Peg13 locus in no.6 is not hypermethylated compared to control. It should be mentioned that the control is Female B6D2F1.

- It appears the Peg13 locus lost some methylation

- The fact that imprinting is mostly maintained, supports the idea that methylation patterns are “normal”

- Introduce “mostly” to tone down this conclusion: The 2 semi-cloned mice possessed mostly normal methylation patterns.

I would personally be curious regarding the methylation status of the oocyte-derived allele of th DMRs. Although not manipulated, showing that their methylation status is as predicted would have supported the “mostly” normal methylation patterns.

2 Lines 275-277

It is not clear to me what is being expressed in this sentence. Can that be rephrased?

Best regards

Jon Schoorlemmer

---

## [Author Response · Author response to Decision Letter 1]

17 Aug 2020

Point to point response to the comments for the authors

1. Lines 223-227.

The description of this Figure is partially incorrect, and the conclusion is still overstated:

- The sentence “patterns ………….appear hypermethylated” is gramatically incorrect.

- The Peg13 locus in no.6 is not hypermethylated compared to control. It should be mentioned that the control is Female B6D2F1.

- It appears the Peg13 locus lost some methylation

- The fact that imprinting is mostly maintained, supports the idea that methylation patterns are “normal”

- Introduce “mostly” to tone down this conclusion: The 2 semi-cloned mice possessed mostly normal methylation patterns.

I would personally be curious regarding the methylation status of the oocyte-derived allele of th DMRs. Although not manipulated, showing that their methylation status is as predicted would have supported the “mostly” normal methylation patterns.

Response: We have corrected the sentence “patterns … appear hypermethylated” and changed the description of DNA methylation at the Peg13. The sentence reads now “Conversely, Peg13 was less methylated in F0 no. 6 and 7 than in the control”

We agree that an analysis of methylation at the H19- or IG-DMR would have allowed us to confirm the unmethylated status of the two DMRs on the oocyte derived chromosomes. Normal development of the two semi-cloned mice is consistent with the expected normal methylation at these 2 imprinted loci. Considering this point, we have toned down our conclusion. The section reads now “Considering that we observed slightly increased methylation levels in the control, which was comparable to that in F0 no. 6 and no. 7, we suggest that methylation of the 3 imprinted genes that we investigated was within a normal range in the 2 semi-cloned mice. This view is consistent with our observation that the semi-cloned mice were healthy and fertile.”

2. Lines 275-277

It is not clear to me what is being expressed in this sentence. Can that be rephrased?

Response: We have rephrased this paragraph. It is harder to explain how mixed diploid and tetraploid cells could have arisen in a single semi-cloned embryo and we have extended the discussion and include two additional references. We hope that the revised paragraph will be informative for the reader.

---

## [Editor Report · Decision Letter 2]

26 Aug 2020

Polyploidy of semi-cloned embryos generated from parthenogenetic haploid embryonic stem cells

PONE-D-20-12064R2

Dear Dr. Wutz,

We’re pleased to inform you that your manuscript has been judged scientifically suitable for publication and will be formally accepted for publication once it meets all outstanding technical requirements.

Kind regards,

Jon Schoorlemmer, PhD

Academic Editor

PLOS ONE

Additional Editor Comments (optional):

Dear author, dear Dr Wutz,

I consider the altered phrasing of high quality and have no further comments. I am happy to inform you that your manuscript "Polyploidy of semi-cloned embryos generated from parthenogenetic haploid embryonic

stem cells" is now acceptable for publication.

---

## [Editor Report · Acceptance letter]

1 Sep 2020

PONE-D-20-12064R2 

Polyploidy of semi-cloned embryos generated from parthenogenetic haploid embryonic stem cells 

Dear Dr. Wutz:

I'm pleased to inform you that your manuscript has been deemed suitable for publication in PLOS ONE. Congratulations! Your manuscript is now with our production department. 

Kind regards, 

on behalf of

Dr. Jon Schoorlemmer 

Academic Editor

PLOS ONE